# M3D: Advancing 3D Medical Image Analysis with Multi-Modal Large Language Models

## Abstract

Medical image analysis is essential to numerous practicals of clinical diagnosis and treatment. However, due to the data scarcity and expensive training cost, previous research has largely focused on 2D medical image analysis, leaving 3D medical images under-explored, despite their important spatial information. This paper aims to advance 3D medical image analysis by leveraging multi-modal large language models (MLLMs). We propose *M3D-LaMed*, a generalist MLLM for 3D medical image analysis, specializing in eight important tasks, including image-text retrieval, report generation, visual question answering, positioning, segmentation, etc. The spatial pooling perceiver is proposed to reduce the 3D tokens, while preserving spatial information. To train the model, we construct the largest 3D multi-modal medical dataset, *M3D-Data*, comprising 120K image-text pairs and 662K instruction-response pairs specifically tailored for 3D medical tasks. The 3D multi-modal benchmark, *M3D-Bench*, is designed, which facilitates the comprehensive evaluation of models across eight tasks. The extensive experiments demonstrate that, as a generalist model, M3D-LaMed shows promising performances and outperforms other specialist models in multiple tasks. With the proposed model, data and benchmark, this work establishes a universal framework that significantly advances the 3D medical image analysis. All data, code and models will be publicly accessible.

## 1 Introduction

Medical practices (Pei et al., 2023) encompass a wealth of multi-modal data that are mainly presented in diagnostic reports and medical images. Paired with medical images, diagnostic reports offer precise description and diagnoses, serving as high-quality annotations. How to effectively leverage such multi-modal data to develop models for generic medical image analysis is a challenging but valuable research topic.

Recent progress in natural image-text understanding (Li et al., 2023b; Liu et al., 2023; Zhu et al., 2023; OpenAI et al., 2023) has highlighted the impressive capabilities of multi-modal large language models (MLLMs) in tasks, such as captioning and visual question answering. They typically integrate vision encoders (Radford et al., 2021; Sun et al., 2023; Zhai et al., 2023) with large language models (LLMs) (Touvron et al., 2023; Zheng et al., 2023; Du et al., 2022; Chowdhery et al., 2022; OpenAI, 2019) and then jointly training on instruction data. As a consequence, MLLMs have garnered much attention of researchers, particularly in medical image analysis. Early works (Zhang et al., 2023; Li et al., 2023a; Wu et al., 2023a; Zhang et al., 2024) have explored building MLLMs for 2D medical tasks, such as report generation and visual question answering on 2D medical images. While these works show promising results, they still struggle with dealing with 3D medical images, such as CT and MRI scans, which are the natural presentation of human body and contain important spatial information of complicated organs and tissues.

In this work, we introduce MLLMs to 3D medical image analysis. Specifically, we propose *M3D-LaMed*, a generalist 3D MLLM specialized in multiple tasks, including image-text retrieval, report generation, and

Table 1: Comparing the constructed M3D-Data with other medical datasets. VQA: Visual Question Answering, ITR: Image-Text Retrieval, RG: Report Generation, REC: Referring Expression Comprehension, REG: Referring Expression Generation, SS: Semantic Segmentation, RES: Referring Expression Segmentation.

| Datasets | Types | Tasks | Images | Texts |
|---|---|---|---|---|
| VQA-Med (Ben Abacha et al., 2019) | 2D | VQA | 3,200 | 12,792 |
| MIMIC-CXR (Johnson et al., 2019) | 2D | ITR, RG | 377,110 | 227,835 |
| PMC-OA (Lin et al., 2023) | 2D | ITR, RG | - | 1,646,592 |
| PMC-VQA (Zhang et al., 2023) | 2D | VQA | 149,075 | 226,946 |
| RP3D-Caption (Wu et al., 2023b) | 3D | ITR, RG | 51K | - |
| RP3D-VQA (Wu et al., 2023b) | 3D | VQA | - | 142K |
| **M3D-Cap** | 3D | ITR, RG | 120,092 | 42,496 |
| **M3D-VQA** | 3D | VQA | 96,170 | 509,755 |
| **M3D-RefSeg** | 3D | REC, REG, SS, RES | 210 | 2,778 |
| **M3D-Seg** | 3D | REC, REG, SS, RES | 5,772 | 149,196* |

* In segmentation datasets, the number of texts can be linked to semantic masks.

visual question answering, along with positioning and segmentation tasks for the first time. Utilizing the 3D vision encoder, pre-trained under the CLIP-like manner (Radford et al., 2021), and the proposed 3D spatial pooling perceiver, M3D-LaMed can effectively process 3D images with less computation. Notably, it integrates with a 3D promptable segmentation model and enables referring expression segmentation of 3D medical images. To train the model, we collect large-scale multi-modal medical data, and then construct the largest public 3D multi-modal medical dataset to date, namely, *M3D-Data*, which comprises 120K image-text pairs and 662K instruction-response pairs covering various diseases and tasks. Furthermore, the first comprehensive benchmark in 3D medical image analysis, *M3D-Bench*, is introduced for evaluating eight 3D medical tasks. Multiple metrics are designed to evaluate models automatically and reliably.

In summary, our contributions are as follows:

- **M3D-LaMed**: A generalist MLLM specialized in various 3D medical tasks, including image-text retrieval, report generation, visual question answering, positioning, segmentation, etc.
- **M3D-Data**: The largest public 3D multi-modal medical dataset to date, with 120K image-text and 662K instruction-response pairs.
- **M3D-Bench**: The first comprehensive benchmark for analyzing model performance on eight distinct 3D multi-modal medical tasks.

## 2 DATASET

We construct M3D-Data to serve as a foundation dataset for supporting a wide range of 3D multi-modal medical tasks. M3D-Data comprises 120K image-text and 662K instruction-response pairs covering 8 tasks, as outlined in Table 1.

### 2.1 IMAGE-TEXT PAIR DATA

Hospitals maintain extensive repositories of medical images and diagnostic reports. However, releasing these image-text data poses significant challenges due to patient privacy concerns. To address this, we sourced medical images and reports from a publicly accessible professional website, Radiopaedia [1]. Each case in our dataset includes multiple 3D images and corresponding reports, along with peer-reviewed captions from

[1]Radiopaedia: https://radiopaedia.org/

Figure 1: The pipelines for generating M3D-Data. (a) In the VQA generation pipeline, the LLM is prompted to generate Q&As based on medical reports. (b) For positioning and segmentation, image-mask-text triplets are created using label-based, definition-based, and annotated instructions. Box coordinates for positioning are derived from the segmentation masks.

Radiopaedia[2] experts. We focus on 3D CT data for its crucial role in diagnosing and measuring lesions. This effort leads to the creation of M3D-Cap, a large-scale dataset comprising 120K 3D medical image-text pairs, which supports tasks such as image-text retrieval and report generation.

## 2.2 INSTRUCTION-RESPONSE PAIR DATA

The instruction-response data includes pairs of instructions or questions and their corresponding responses. This data is important for training models to implement multi-modal tasks such as Visual Question Answering (VQA), positioning, and segmentation, totaling 662K instruction-response pairs.

**VQA Data**: Acquiring medical VQA data is costly due to the need for expert involvement. To reduce expenses, we employed public LLMs to analyze text reports and generate instruction-response pairs using a prompt-based approach (Figure 1(a)). We applied self-filtering techniques to eliminate noisy data, with 7K samples reviewed by 10 experts, resulting in a pass rate exceeding 95% (Table 9). Our findings indicate that a powerful open-source model can efficiently and economically generate accurate Q&A pairs from medical reports. Therefore, we utilized the Qwen-72B model (Bai et al., 2023) instead of ChatGPT (OpenAI, 2019), creating multiple-choice Q&As on five key topics: imaging plane, imaging phase, organ, abnormality, and location (Figure 4), facilitating both open- and closed-ended evaluations.

**Positioning and Segmentation Data**: Positioning and segmentation tasks require integrating images, text, and referring regions, typically as bounding boxes or segmentation masks. We simplify data handling with a unified format of image-mask-text triplets, converting masks to 3D box coordinates for positioning tasks. Given the scarcity of lesion mask annotations in clinical, creating a 3D image-mask-text dataset is resource-

intensive. To address this, we use three methods (Figure 1(b)): **(1) Label-based instruction:** Generated from public segmentation datasets using label templates. **(2) Definition-based instruction:** Built using a term dictionary and LLM-generated definitions. **(3) Annotated instruction:** Created via expert-annotated text descriptions referring to specific regions. We use the Qwen-72B model to augment instructions. Methods (1) and (2) compile the M3D-Seg dataset from public 3D CT segmentation data (see Appendix), while method (3) annotates the M3D-RefSeg dataset from the Totalsegmentator dataset (Wasserthal et al., 2023).

## 2.3 DATA QUALITY

The quality of M3D-Data is rigorously controlled across its four sub-datasets. **M3D-Cap**: Image-text pairs are sourced from Radiopaedia[2], with peer-reviewed cases by their Editorial Board[2]. **M3D-VQA**: Question-answer pairs are derived from M3D-Cap reports. Ten experts reviewed a sample of 7K data points, covering five question types (plane, phase, organ, abnormality, location) across three splits (train: 1K, validation: 1K, test: 5K). All test data are expert-reviewed, yielding an average pass rate of over 95% (Table 9). Experts correct the test set, which will serve as a benchmark. Detailed quality analysis is provided in the appendix. **M3D-Seg**: Includes 25 public segmentation datasets, all validated through publications or challenges. The detailed dataset list is in the appendix. **M3D-RefSeg**: Based on the TotalSegmentator dataset, experts cross-validated textual descriptions for the image masks.

## 3 METHOD

Firstly, we pre-train the 3D medical vision encoder under a CLIP-like manner (Radford et al., 2021) on the M3D-Cap dataset (Figure 2(a)). Then, the spatial pooling perceiver is tuned for feature alignment using image-text pairs in M3D-Data, with vision encoder and LLM frozen. Finally, we perform instruction tuning on all modules, enabling smooth integration of vision and language modules (Figure 2(b)).

### 3.1 MODEL ARCHITECTURE

**3D Image Encoder**: For a given 3D image $I \in \mathbb{R}^{C \times D \times H \times W}$, where $C, D, H,$ and $W$ represent channels, depth, height, and width, the image embedding is computed as $v = E_{img}(I) \in \mathbb{R}^{n \times d}$, where $E_{img}$ is the image encoder, $n$ is the number of image tokens, and $d$ is the token dimensions. We use the 3D Vision Transformer (Dosovitskiy et al., 2021) as the encoder, which processes image patches of size $P_D * P_H * P_W$ through its $N$-layer transformer.

**3D Perceiver**: To reduce the high computational costs of processing 3D images with LLMs, we propose an efficient 3D spatial pooling perceiver. This module reduces the number of visual tokens and then projects them to the same embedding dimension as LLM (Figure 2(c)). Specifically, the vision encoder's output tokens are reconstructed into 3D shape for pooling, reducing token number while preserving spatial information. Then, a series of Multi-Layer Perceptrons (MLPs) adjust embedding dimensions to match the LLM's input requirements. This approach reduces computational load while retaining essential spatial features.

**LLM**: Large language models (LLMs) trained on vast natural language corpora provide versatile embeddings and strong generative capabilities. M3D-LaMed can easily integrate with any advanced LLM. We evaluate several efficient and high-performing LLMs, including Llama-2-7B (Touvron et al., 2023), Llama-3-8B (AI@Meta, 2024), and Phi3-4B (Abdin et al., 2024), which excel at capturing linguistic patterns and generating coherent text across various domains.

**Promptable Segmentation Module**: Inspired by LISA (Lai et al., 2023), we utilize MLLMs for referring expression segmentation via a promptable segmentation module. When a [SEG] token appears in the output,

---

[2]Editorial Board: https://radiopaedia.org/editors

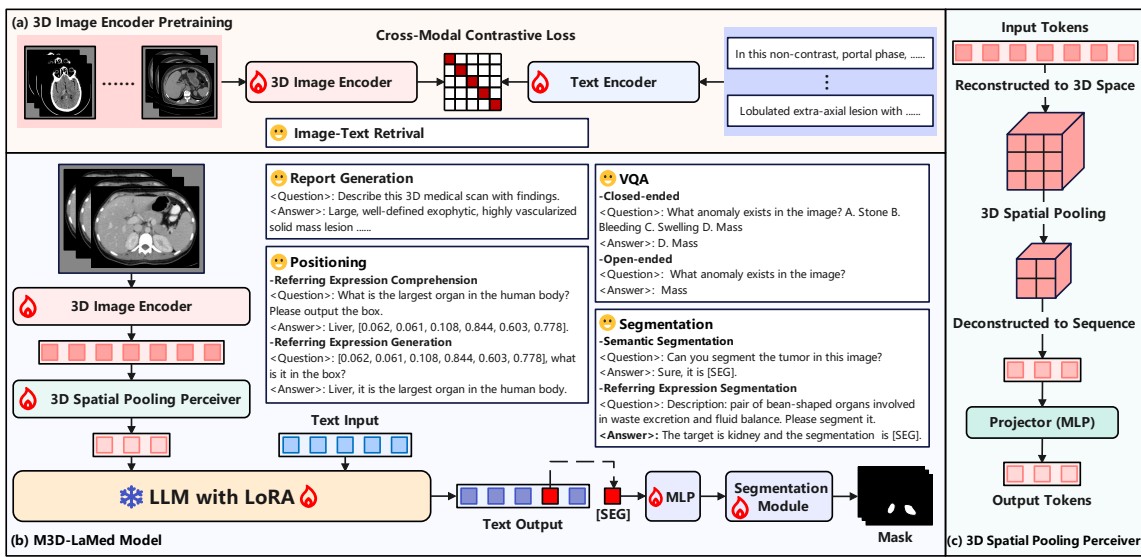

Figure 2: Overview of the M3D-LaMed model. (a) The 3D image encoder is pre-trained with cross-modal contrastive learning on M3D-Cap data. (b) For inference, 3D medical images are processed by the 3D image encoder and then 3D spatial pooling perceiver. Later, the tokens are injected into the LLM. The [SEG] token prompts the 3D medical segmentation model to generate the corresponding 3D mask. Powered by the M3D-Data training data, M3D-LaMed supports diverse 3D medical tasks. (c) The details of 3D spatial pooling perceiver: It reconstructs the 3D shape from the input token sequence for spatial pooling and reducing token count, then deconstructs them back into token sequence. The projection layer with MLPs adjusts the token dimensions to match the language dimensions in LLM.

we extract its last-layer embedding and map it into a prompt. This prompt drives the segmentation module through MLPs to generate the segmentation mask. We selected SegVol (Du et al., 2023) as the promptable segmentation module for its robust performance and compatibility with our framework.

## 3.2 MODEL TRAINING

**Setup**: We preprocess the 3D CT images using Min-Max Normalization, followed by resizing and cropping to a standard dimension of $32 \times 256 \times 256$. Our 3D vision encoder employs a 3D ViT with 12 layers and a patch size of $4 \times 16 \times 16$, yielding output embeddings of $2048 \times 768$, representing 2048 tokens with 768 feature dimensions each. After applying the 3D spatial pooling perceiver, the final vision tokens fed to the LLM are $256 \times 768$. All models are trained by AdamW optimizer (Kingma & Ba, 2014; Loshchilov & Hutter, 2017) with warm-up and cosine decay, and use the bf16 mixed-precision training strategy enabled by DeepSpeed. Training is conducted in parallel across 8 NVIDIA A100 GPUs (80 GB each).

**Vison Encoder Pre-training**: To address the lack of robust 3D medical image encoders, we adopt the CLIP (Radford et al., 2021) architecture and training methodology to pre-train on the M3D-Cap dataset using cross-modal contrastive learning loss (Figure 2(a)). The vision encoder is trained from scratch, while the text encoder is initialized using a pre-trained BERT model (Devlin et al., 2019), which consists of 12 transformer layers and accommodates a maximum text length of 128 tokens. Both encoders use $[CLS]$ tokens for global

feature representation, and a linear layer projects these representations into a suitable space for contrastive training. We use a batch size of $32 \times 8$ for parallel training across 8 GPUs, with a learning rate of $10^{-4}$.

**MLLM Feature Alignment**: We first freeze both the vision encoder and the LLM, fine-tuning only the 3D perceiver to align the vision and language models with image-text pairs from M3D-Cap and M3D-VQA. This process utilizes a batch size of $16 \times 8$ and a learning rate of $10^{-4}$.

**MLLM Instruction Tuning**: We fine-tune the vision encoder, 3D perceiver, LLM, and segmentation module using the complete M3D-Data. When the [SEG] token appears in the output, we apply Dice loss and Binary Cross-Entropy (BCE) loss for segmentation training. To control training costs while preserving the LLM's original knowledge, we utilize the LoRA strategy (Hu et al., 2021) for parameter-efficient fine-tuning, using a batch size of $8 \times 8$ and a learning rate of $2 \times 10^{-5}$. We set LoRA parameters to $r = 16$, $\alpha = 32$, and a dropout rate of 0.05, with a maximum context length of 512 tokens. The segmentation module initializes with parameters from SegVol (Du et al., 2023).

## 4 BENCHMARK AND EVALUATION

It lacks proper benchmark for evaluating 3D multi-modal medical tasks. Thus, we construct *M3D-Bench* for comprehensive evaluation of models across eight tasks which are categorized into five key abilities: image-text retrieval, report generation, VQA, positioning, and segmentation.

**Benchmarking Image-Text Retrieval.** In 3D image-text retrieval, the goal is to match images and texts based on their similarity, involving two sub-tasks: text-to-image retrieval (TR) and image-to-text retrieval (IR). For evaluation, we utilize a high-quality subset of 2,000 pairs from M3D-Cap as the test set. This set is stratified into four difficulty levels—easy (100 pairs), medium (500 pairs), difficult (1,000 pairs), and very difficult (2,000 pairs)—based on the size of the retrieval candidate pool. Evaluation metrics include recall at ranks 1, 5, and 10 for both IR and TR, which assess the model's ability to retrieve relevant images or texts among the top-ranked results.

**Benchmarking Report Generation.** In report generation, the model generates text reports based on information extracted from 3D medical images. We evaluate performance using a test set of 1,000 image-text pairs for user assessment. Given the complexity of evaluating content accuracy between generated reports and human references, we employ both traditional and LLM-based metrics.

Traditional metrics include BLEU (Papineni et al., 2002), ROUGE (Lin, 2004), METEOR (Banerjee & Lavie, 2005), and BERT-Score (Zhang et al., 2019), which quantify text similarity through n-gram overlap and variations, although they have limited semantic understanding.

LLM-based metrics, such as the GREEN (Ostmeier et al., 2024) score, utilize models with strong semantic comprehension to evaluate the alignment between generated reports and human references. This metric assesses matching content and errors, offering a more comprehensive measure of report quality.

**Benchmarking VQA.** The VQA tasks involve generating text-based answers in response to images and questions, categorized into open-ended and closed-ended formats. Open-ended VQA allows unrestricted answer generation, while closed-ended VQA limits responses to a predefined set of choices.

We organize M3D-VQA as multiple-choice questions with four possible answers (A, B, C, D). Two test sets are provided: the basic test set, which includes 2,000 3D medical images and 13,791 question-answer pairs across five question types, and the small test set, comprising 1,000 images and 5,000 pairs, also covering the same five types. Results are based on the basic test set, while the small test set facilitates quicker evaluations. After self-filtering to remove low-quality data, expert reviews ensured a pass rate of 96.3%.

For closed-ended VQA, accuracy is assessed by the model's ability to match answers to provided choices. In open-ended VQA, the evaluation involves comparing generated answers to reference answers using metrics such as BLEU, ROUGE, METEOR, and BERT-Score.

**Benchmarking Positioning.** Positioning is essential in vision-language tasks (Chen et al., 2023), especially those involving input and output boxes. For tasks with output boxes, such as Referring Expression Comprehension (REC) (Kazemzadeh et al., 2014; Mao et al., 2016), the goal is to localize a target object in an image based on a referring expression. Conversely, tasks with input boxes, like Referring Expression Generation (REG) (Liu et al., 2017), require the model to describe a specific region given an image and a location box.

In our datasets, M3D-RefSeg and M3D-Seg, masks are converted into box coordinates representing the maximum bounding rectangle $(x_1, y_1, z_1, x_2, y_2, z_2)$. For evaluation, 20% of the data from AbdomenCT-1K (Ma et al., 2022) within M3D-Seg is utilized as the test set. Positioning performance for output boxes is assessed using the Intersection over Union (IoU) metric, while the quality of generated descriptions for input boxes is evaluated with BLEU, ROUGE, METEOR, and BERT-Score.

**Benchmarking Segmentation.** Segmentation is vital for 3D medical image analysis, enabling recognition and localization. It is divided into semantic segmentation, where models generate masks based on predefined semantic labels, and referring expression segmentation, which segments targets described by natural language.

For evaluation, 20% of the data from AbdomenCT-1K (Ma et al., 2022), TotalSegmentator (Wasserthal et al., 2023), and CT-Organ (Rister et al., 2020) in the M3D-Seg is designated as the test set for both segmentation types. The Dice is used as the primary evaluation metric for these tasks.

## 5 EXPERIMENTS

**Experiments on Image-Text Retrieval.** Given the lack of suitable CLIP-like models for 3D medical image analysis, we use a 2D medical model as a baseline. We evaluate the 2D model by sampling 10 equally spaced 2D slices from each 3D image along the depth dimension, identifying the 3D image with the slice that shows the highest similarity to the target. Although we initially considered using CLIP, it yielded poor results in the medical domain, prompting us to select PMC-CLIP (Lin et al., 2023) as our baseline. As shown in Table 2, our model significantly outperforms the PMC-CLIP model across various difficulty levels, primarily due to PMC-CLIP's limited spatial information. In the easiest setting (100 test samples, R@10), our model achieves a 55% improvement in image-to-text retrieval (IR). In the most challenging setting (2000 samples, R@1), our model exceeds PMC-CLIP by 77.40%. We also examined the effect of training batch size on performance, finding that larger batch size yield significant gains. Specifically, increasing the batch size from 6 to 32 results in a 59.45% improvement in the most difficult setting (2000 samples, R@1).

**Experiments on Report Generation.** Table 3 compares the performance of the RadFM and M3D-LaMed models across five metrics. Leveraging the large-scale M3D dataset and the 3D perceiver, M3D-LaMed outperforms RadFM in all metrics, with the LaMed-Phi-3-4B model exceeding RadFM by 23.48% in GREEN scores. Among the M3D-LaMed models, the Phi-3-4B-based model achieves the highest performance, aligning with the overall ranking of the underlying LLMs (Abdin et al., 2024). Notably, the Phi-3-4B model consistently outperforms the Llama-2-7B and Llama-3-8B models across most language benchmarks, despite having fewer parameters, demonstrating its superior pre-training language capabilities.

**Experiments on VQA.** We evaluated the performance of our M3D-LaMed models and RadFM on closed-ended and open-ended VQA tasks. Table 4 shows that our model significantly outperforms RadFM across all

Table 2: Comparison of image-text retrieval performance. Our model outperforms previous models across various difficulty levels, with larger batch sizes further enhancing performance. IR (image-to-text retrieval), TR (text-to-image retrieval). Metrics R@1, R@5, and R@10 represent recall rates at ranks 1, 5, and 10.

| Methods | | PMC-CLIP (Lin et al., 2023) | | | | Our (Batch Size: 6) | | | | Our (Batch Size: 32) | | | |
|---|---|---|---|---|---|---|---|---|---|---|---|---|---|
| Test Samples | | 100 | 500 | 1000 | 2000 | 100 | 500 | 1000 | 2000 | 100 | 500 | 1000 | 2000 |
| IR | R@1 | 9.00 | 4.40 | 1.90 | 1.15 | 64.00 | 39.60 | 27.30 | 19.10 | 95.00 | 86.20 | 82.20 | 78.55 |
| | R@5 | 28.00 | 12.80 | 7.60 | 4.35 | 95.00 | 76.20 | 61.10 | 47.45 | 99.00 | 96.80 | 95.00 | 93.20 |
| | R@10 | 45.00 | 18.80 | 12.10 | 7.60 | 99.00 | 87.20 | 76.10 | 62.25 | 100.00 | 97.80 | 97.20 | 95.75 |
| TR | R@1 | 18.00 | 7.60 | 4.60 | 3.15 | 70.00 | 40.40 | 26.60 | 18.45 | 94.00 | 86.20 | 81.70 | 77.95 |
| | R@5 | 47.00 | 20.20 | 13.00 | 8.55 | 95.00 | 74.20 | 61.80 | 47.30 | 100.00 | 96.40 | 94.60 | 93.40 |
| | R@10 | 59.00 | 31.00 | 19.80 | 13.55 | 98.00 | 87.00 | 75.30 | 62.15 | 100.00 | 97.40 | 96.90 | 96.25 |

Table 3: Comparison on report generation.

| Methods | BLEU | ROUGE | METEOR | BERT-Score | GREEN |
|---|---|---|---|---|---|
| RadFM-14B (Wu et al., 2023b) | 12.23 | 16.49 | 11.57 | **87.93** | 3.98 |
| LaMed-Llama-2-7B | 18.96 | 23.11 | 17.54 | 84.32 | 6.79 |
| LaMed-Llama-3-8B | 29.50 | 33.18 | 28.39 | 86.43 | 19.50 |
| LaMed-Phi-3-4B | **36.19** | **39.78** | **35.24** | 87.70 | **27.46** |

five question types for closed-ended VQA, primarily due to the larger M3D dataset, which is approximately four times the size of the RP3D-VQA dataset. When comparing different LLM bases, the LaMed-Phi-3-4B model exceeds the Llama-based models by 3.66% and 3.27% in mean scores, reflecting Phi-3-4B's robust language knowledge, which enhances its performance in closed-set VQA tasks. For open-ended VQA, Table 5 indicates that our model again significantly outperforms RadFM across all question types and evaluation metrics, attributed once more to the larger M3D dataset. Among the M3D-LaMed series, the LaMed-Llama-3-8B model performs best, surpassing LaMed-Llama-2-7B by 0.98% and LaMed-Phi-3-4B by 1.58% in BLEU scores, primarily due to the size of the model parameters.

**Experiments on Positioning.** Figure 3 evaluates the 3D positioning task, which includes two subtasks: Referring Expression Comprehension (REC) for output with a bounding box and Referring Expression Generation (REG) for input with a bounding box. Freezing the visual encoder during training significantly reduces positioning performance, resulting in a 33.19% decrease in IoU score for REC and a 26.87% drop in the BLEU score for REG. Among the M3D-LaMed models, LaMed-Phi-3-4B demonstrates superior performance, especially in REC, where its IoU scores exceed those of other models by 3.1% and 4.78%. This enhanced performance is likely due to the robust pre-training of the Phi-3-4B LLM.

Table 4: Comparison on 3D closed-ended VQA in five types of questions.

| Methods | Plane | Phase | Organ | Abnormality | Location | Mean |
|---|---|---|---|---|---|---|
| RadFM-14B (Wu et al., 2023b) | 19.65 | 28.70 | 16.80 | 18.92 | 14.88 | 19.79 |
| LaMed-Llama-2-7B | **99.05** | 88.80 | 79.15 | 71.14 | 65.14 | 80.66 |
| LaMed-Llama-3-8B | 99.00 | 87.95 | 80.30 | 72.57 | 65.44 | 81.05 |
| LaMed-Phi-3-4B | 98.60 | **89.35** | **85.24** | **77.95** | **70.57** | **84.32** |

Table 5: Evaluation on 3D open-ended VQA in five types of questions and four metric evaluations.

| Methods | Metric | Plane | Phase | Organ | Abnormality | Location | Mean |
|---|---|---|---|---|---|---|---|
| RadFM-14B (Wu et al., 2023b) | BLEU | 14.24 | 14.25 | 14.24 | 15.64 | 23.58 | 16.39 |
| | ROUGE | 25.40 | 25.41 | 25.38 | 25.38 | 29.09 | 26.13 |
| | METEOR | 20.62 | 20.63 | 20.61 | **20.60** | **24.19** | 21.33 |
| | BERT-Score | 92.68 | 92.04 | 86.79 | 85.84 | 86.26 | 88.72 |
| LaMed-Llama-2-7B | BLEU | 98.85 | **81.93** | 41.69 | 22.12 | 26.56 | 54.23 |
| | ROUGE | 98.88 | 85.96 | 46.05 | 26.22 | 31.26 | 57.67 |
| | METEOR | 49.44 | 70.73 | 28.91 | 18.41 | 21.07 | 37.71 |
| | BERT-Score | 99.83 | 96.94 | 90.76 | 86.83 | 88.19 | 92.51 |
| LaMed-Llama-3-8B | BLEU | **98.96** | 81.50 | **43.14** | **23.75** | **28.69** | **55.21** |
| | ROUGE | **98.99** | 85.76 | **47.61** | **28.32** | **33.18** | **58.77** |
| | METEOR | **49.51** | 70.37 | **30.04** | 19.72 | 22.61 | **38.45** |
| | BERT-Score | **99.84** | 96.86 | **91.01** | **87.19** | **88.53** | **92.69** |
| LaMed-Phi-3-4B | BLEU | 98.63 | 81.32 | 41.60 | 20.68 | 25.92 | 53.63 |
| | ROUGE | 98.67 | **86.32** | 46.07 | 24.70 | 30.39 | 57.23 |
| | METEOR | 49.36 | **71.32** | 29.31 | 17.51 | 20.75 | 37.65 |
| | BERT-Score | 99.80 | **96.96** | 90.56 | 86.48 | 88.10 | 92.38 |

Table 6: Comparison of 3D segmentation performance. Our model outperforms prior methods in semantic segmentation and performs previously unattainable Referring Expression Segmentation (RES) tasks. The datasets used for comparison include ACT-1K (AbdomenCT-1K (Ma et al., 2022)), TS (TotalSegmentator (Wasserthal et al., 2023)), and CTOrg (CT-Organ (Rister et al., 2020)).

| Methods | Semantic Segmentation | | | | Referring Expression Segmentation | | | |
|---|---|---|---|---|---|---|---|---|
| | ACT-1K | TS | CTOrg | Mean | ACT-1K | TS | CTOrg | Mean |
| SegVol (Du et al., 2023) | 79.06 | 44.28 | 77.78 | 67.04 | - | - | - | - |
| LaMed-Llama-2-7B | 90.18 | **68.58** | 81.86 | **80.21** | 90.18 | 65.83 | **82.91** | **79.64** |
| LaMed-Llama-3-8B | **90.43** | 64.89 | **82.19** | 79.17 | **90.43** | 65.89 | 82.19 | 79.50 |
| LaMed-Phi-3-4B | 89.42 | 64.96 | 81.21 | 78.53 | 89.53 | 62.42 | 80.17 | 77.37 |

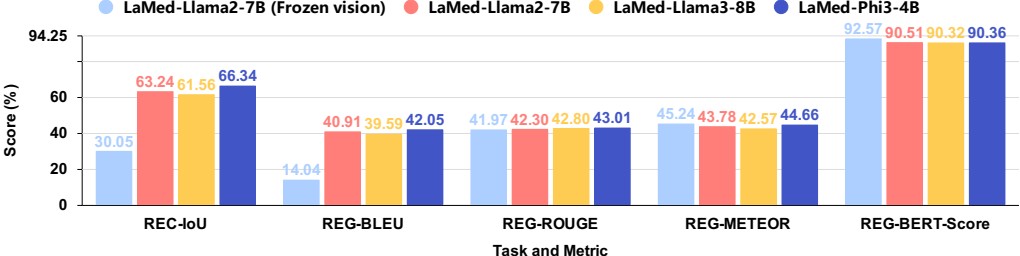

Figure 3: Evaluation on 3D positioning (REC & REG) with M3D-LaMed models. Freezing the vision encoder severely impairs performance on the visual-language positioning task.

**Experiments on Segmentation.** Table 6 evaluates the 3D segmentation task, covering both Semantic Segmentation (SS) and Referring Expression Segmentation (RES). Our models, utilizing the advanced capabilities of MLLMs, outperform SegVol by 13.17% in the mean Dice score across three SS tasks. Additionally, our models provide RES ability, which SegVol lacks. Among the M3D-LaMed series, LaMed-Llama-2-7B

Table 7: Ablation study of the LaMed-Llama-2-7B model for closed-ended VQA.

| Vision Pre-train | Spatial Pooling | MLP | Unlocked Vision | VQA Mean |
|---|---|---|---|---|
| ✘ | ✔ | ✔ | ✘ | 71.13 |
| ✔ | ✘ | ✔ | ✘ | 72.87 |
| ✔ | ✔ | ✘ | ✘ | 73.50 |
| ✔ | ✔ | ✔ | ✘ | 74.96 |
| ✔ | ✔ | ✔ | ✔ | 75.78 |

achieves the best performance, closely followed by LaMed-Llama-3-8B. However, LaMed-Phi-3-4B scores 1.68% lower in SS and 2.27% lower in RES, likely due to its smaller parameter count. The limited impact of powerful pre-trained language models on segmentation tasks helps explain why LaMed-Phi-3-4B does not outperform the others. LaMed-Llama-2-7B and LaMed-Llama-3-8B exhibit comparable performance across various datasets, each showcasing unique strengths.

**Ablation Study.** Table 7 summarizes ablation studies of our LaMed-Llama-2-7B model on closed-set VQA tasks, focusing on four key modules: vision pre-training, spatial pooling, MLP, and unlocked vision. Specifically, removing vision pre-training means training from scratch, resulting in a performance decrease of 3.83, highlighting its critical importance. Additionally, omitting spatial pooling involves directly pooling sequence tokens, leading to a reduction of 2.09. Then, excluding the MLP for a single linear layer decreases performance by 1.46. Consequently, our 3D spatial pooling perceiver employs 3D spatial pooling for token downsampling and the MLP as a projector. Furthermore, the omission of unlocked vision reflects freezing the vision encoder during fine-tuning, resulting in a decline of 0.82. Overall, vision pre-training is the most impactful factor for enhancing performance. Our findings emphasize the significance of each component, with optimal training requiring visual pre-training and an unlocked vision encoder during fine-tuning.

**More Details and Experiments.** We provide more details and experimental results in the appendix. Here is a brief summarization. The detailed discussion about related work is presented in Section A. More details about data distribution and quality can be found in Section B. Section C provides detailed parameters of each module in M3D-LaMed. Section D presents qualitative analysis across tasks of image-text retrieval, report generation, VQA, positioning, and segmentation. The results show that our model outperforms RadFM (Wu et al., 2023b) and GPT-4V (OpenAI et al., 2023). We test the out-of-distribution (OOD) generalization performance in Section E, and our M3D-LaMed can still answer OOD questions reasonably. All prompts and templates used in data construction and experiments, including data generation, task instructions, and term dictionary, are detailed in Appendix Section F.

## 6 CONCLUSION

This work introduces the generalist MLLM *M3D-LaMed*, the largest dataset *M3D-Data*, and the comprehensive benchmark *M3D-Bench* for 3D medical image analysis. We explore the integration of 3D vision encoder, 3D spatial pooling perceiver and LLM in M3D-LaMed. Extensive experiments show that our generalist M3D-LaMed achieves promising results and outperforms other specialist models in corresponding tasks. We believe the contributed model, data and benchmark will facilitate the research of 3D medical image analysis and further clinical practices.

**Limitations.** Despite M3D-LaMed shows remarkable performance in 3D medical analysis, challenges remain in the analysis of higher-resolution and multiple 3D scans, which requires more efficient and lightweight LLMs for processing long token sequences. Although a large-scale 3D medical dataset is contributed, more data needs to be collected and annotated for training better models.

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

## A    RELATED WORK

**Medical Multi-Modal Datasets**: In medical scenarios (Pei et al., 2023), rich images of various modalities and texts are available. However, previous works (Ben Abacha et al., 2019; Johnson et al., 2019) have difficulty constructing large-scale medical multi-modal datasets due to privacy and restrictions. Inspired by CLIP (Radford et al., 2021), PMC-OA (Lin et al., 2023) obtained image and text data from medical papers through web crawling, resulting in 1.6M 2D image-text pairs. Additionally, MedMD (Wu et al., 2023b) aims to achieve multiple objectives: building 2D and 3D medical models, integrating public 2D medical datasets, and crawling 3D image and text data from medical professional websites. One of its 3D datasets, RP3D (Wu et al., 2023b), comprises 51K 3D image-text pairs and 142K VQA data generated from LLMs. In our work, we primarily focus on constructing large-scale 3D medical datasets by crawling medical professional websites. M3D-Data includes 120K 3D image-text pairs and 662K instruction-response pairs generated through an automatic and low-cost data generation pipeline. Furthermore, M3D-Data's M3D-Seg component collects nearly 6K 3D images from 25 public medical segmentation datasets, facilitating tasks such as positioning and segmentation. In summary, M3D-Data is the largest 3D medical multi-modal dataset, supporting various tasks, as shown in Table 1.

**Medical MLLMs**: Medical MLLMs (Li et al., 2023a; Wu et al., 2023a; Zhang et al., 2024) are typically fine-tuned from powerful 2D open-source MLLMs using medical multi-modal datasets. For instance, LLaVA-Med (Li et al., 2024), Med-PaLM M (Tu et al., 2024), and Med-Flamingo (Moor et al., 2023) are based on models such as LLaVA (Liu et al., 2023), PaLM-E (Driess et al., 2023), and Flamingo (Alayrac et al., 2022), respectively. The availability of large-scale datasets like PMC-VQA (Zhang et al., 2023) has enabled training medical MLLMs from scratch, although initially limited to 2D images. While RadFM (Wu et al., 2023b) supports both 2D and 3D images, it is primarily used for 2D images and text generation tasks such as VQA and performs poorly on 3D images. In our work, M3D-LaMed serves as a generalist MLLM for 3D medical image analysis. It handles not only text generation tasks like report generation and VQA but also pioneers vision tasks, like positioning and segmentation in 3D medical images, which are crucial for identification and localization in medical image analysis.

## B    DATASETS

**Data Statistics for M3D-VQA**: We analyzed the data distribution in the M3D-VQA dataset, as shown in Figure 4. The most frequent initial words are "what," "which," and "where," indicating a balanced distribution of question types. We also calculate the proportions of the five question types and illustrate the answer samples using a word cloud in Figure 4.

**Data List for M3D-Seg**: In addition to the detailed introduction of our datasets in Section 3 of the manuscript, we provide additional details of the M3D-Seg dataset in Table 8. M3D-Seg comprises 5,772 3D CTs and their corresponding masks collected from 25 public segmentation datasets. These datasets offer training and evaluation data for positioning and segmentation tasks.

**Data Quality**: Question-answer pairs in this dataset are generated based on the reports from M3D-Cap. We employed 10 qualified experts to review a sample of 7K data points, which includes five types of questions (Q1: plane, Q2: phase, Q3: organ, Q4: abnormality, Q5: location) across three data splits (train: 1K, validation: 1K, test: 5K). Specifically, all test data were reviewed and corrected by experts, and the revised test set will be made public as a benchmark. As shown in Table 9, the average passing rate exceeds 95%, confirming the dataset's high quality. In addition, we identified three types of errors during data validation: *Hallucination*, content is imagined that does not exist in the report; *Non-unique*, the answer is not unique; *Others*, miscellaneous errors. We observed that Q3, related to multiple organs, is prone to non-unique errors, while Q4 and Q5, focused on abnormalities and locations, tend to have hallucination errors. To address these issues, we engaged 10 experts to manually correct all test data.

Table 8: Detailed dataset composition in M3D-Seg. M3D-Seg contains 5,772 labeled 3D CTs from 25 public datasets. All data, download links, and processing scripts will be made public. CAT: Category. 3D-IRCADB (Soler et al., 2010), FLARE22 (Ma et al., 2023), AbdomenCT-1k (Ma et al., 2022), AMOS22 (Ji et al., 2022), BTCV (Landman et al., 2015), CHAOS (Kavur et al., 2021; 2019; 2020), CT-ORG (Rister et al., 2019; 2018; Bilic et al., 2023; Clark et al., 2013), HaN-Seg (Podobnik et al., 2023), KiPA22 (He et al., 2021; 2020; Shao et al., 2011; 2012), KiTS19 (Heller et al., 2020), KiTS23 (Heller et al., 2023), LUNA16 (Setio et al., 2017), MSD-Colon (Simpson et al., 2019), MSD-HepaticVessel (Simpson et al., 2019), MSD-Liver (Simpson et al., 2019), MSD-Lung (Simpson et al., 2019), MSD-Pancreas (Simpson et al., 2019), MSD-Spleen (Simpson et al., 2019), Pancreas-CT (Roth et al., 2016; 2015; Clark et al., 2013), QUBIQ (QUB), SLIVER07 (Heimann et al., 2009), TotalSegmentator (Wasserthal et al., 2022), VerSe19 (Sekuboyina et al., 2021; Löffler et al., 2020; Liebl et al., 2021), VerSe20 (Sekuboyina et al., 2021; Löffler et al., 2020; Liebl et al., 2021), WORD (Luo et al., 2022).

| Datasets | Anatomical Targets | CAT | Train | Test | All |
|---|---|---|---|---|---|
| 3D-IRCADB | Liver and liver tumor | 47 | 16 | 4 | 20 |
| FLARE22 | Thoracic and abdominal organs | 13 | 40 | 10 | 50 |
| AbdomenCT-1k | Liver, kidney, spleen, pancreas | 4 | 800 | 200 | 1000 |
| AMOS22 | Abdominal organs | 15 | 192 | 48 | 240 |
| BTCV | Abdominal organs | 13 | 24 | 6 | 30 |
| CHAOS | Abdominal organs | 1 | 16 | 4 | 20 |
| CT-ORG | Organs of the body | 6 | 112 | 28 | 140 |
| HaN-Seg | Organs of the head and neck | 30 | 33 | 9 | 42 |
| KiPA22 | Kidney, renal tumor, artery, vein | 4 | 56 | 14 | 70 |
| KiTS19 | Kidney and kidney tumor | 2 | 168 | 42 | 210 |
| KiTS23 | Kidney, kidney tumor and cyst | 3 | 391 | 98 | 489 |
| LUNA16 | Left lung, right lung, trachea | 3 | 710 | 178 | 888 |
| MSD-Colon | Colon tumor | 1 | 100 | 26 | 126 |
| MSD-HepaticVessel | Hepatic vessel and liver tumor | 2 | 242 | 61 | 303 |
| MSD-Liver | Liver and liver tumor | 2 | 104 | 27 | 131 |
| MSD-Lung | Lung tumor | 1 | 50 | 13 | 63 |
| MSD-Pancreas | Pancreas and pancreas tumor | 2 | 224 | 57 | 281 |
| MSD-Spleen | Spleen | 1 | 32 | 9 | 41 |
| Pancreas-CT | Pancreas | 1 | 65 | 17 | 82 |
| QUBIQ | Kidney, pancreas and lesion | 3 | 65 | 17 | 82 |
| SLIVER07 | Liver | 1 | 16 | 4 | 20 |
| TotalSegmentator | Organs of the whole body | 104 | 962 | 241 | 1203 |
| VerSe19 | Vertebrae | 28 | 64 | 16 | 80 |
| VerSe20 | Vertebrae | 28 | 48 | 13 | 61 |
| WORD | Thoracic and abdominal organs | 16 | 80 | 20 | 100 |
| **Sum** | - | - | **4610** | **1162** | **5772** |

Table 9: The pass rate (%) of expert examination on M3D-VQA. Error type ratio (Hallucination : Non-unique : Others). Note that we have organized 10 experts to correct all validation and test data.

| Split | Q1 | Q2 | Q3 | Q4 | Q5 | Avg. |
|---|---|---|---|---|---|---|
| Train | 100 | 98.5 | 97.0 | 92.0 | 90.5 | 95.6 |
| Val | 100 | 100 | 98.5 | 91.5 | 91.5 | 96.3 |
| Test | 100 | 99.8 | 98.0 | 95.9 | 91.1 | 97.0 |
| H:N:O | - | 0:0:10 | 2:6:2 | 8:1:1 | 6:2:2 | 5:3:2 |

Table 10: The parameters of each module in our M3D-LaMed. We utilize 3D ViT with a 12-layer transformer as a 3D image encoder, Llama-2-7B (Touvron et al., 2023), Llama-3-8B (AI@Meta, 2024), and Phi-3-4B (Abdin et al., 2024) as LLM bases, and SegVol (Du et al., 2023) as a segmentation module.

| Modules | Parameters |
|---|---|
| 3D Image Encoder | 87.4M |
| 3D Spatial Pooling Perceiver | 19.9M |
| LLM with LoRA (Llama-2-7B / Llama-3-8B / Phi-3-4B) | 6.7B / 8.1B / 3.8B |
| Segmentation Module | 117.3M |
| All | 6.9B / 8.3B / 4.0B |

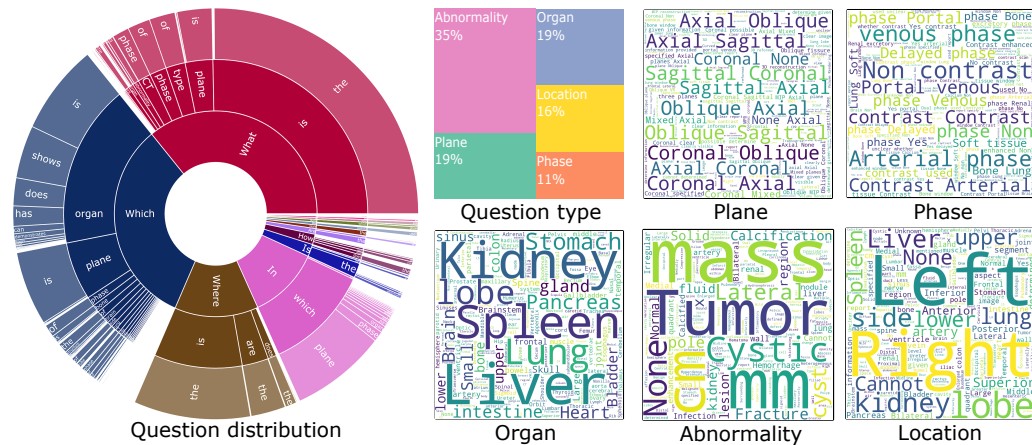

Figure 4: The data statistics for M3D-VQA are categorized into five question types, with "what," "which," and "where" being the three most common question types. Word clouds are used to visualize sample distributions across the five topics.

## C MODEL PARAMETERS

Detailed module parameters of the M3D-LaMed model are presented in Table 10. Specifically, we explore three LLM bases: Llama-2-7B (Touvron et al., 2023), Llama-3-8B (AI@Meta, 2024), and Phi-3-4B (Abdin et al., 2024). The overall model parameters amount from 4.0 to 8.3 billion, considerably smaller than RadFM (Wu et al., 2023b), which has 14 billion parameters. Although the LLM base constitutes 97% of all parameters, fine-tuning LLM with just LoRA during training is exceptionally cost-effective.

## D QUALITATIVE ANALYSIS

To further demonstrate our model's performance and generalist ability on 3D multi-modal medical tasks, we add qualitative analysis on 8 tasks: image-text retrieval (Figure 5), report generation (Figure 6), closed-ended VQA (Figure 7), open-ended VQA (Figure 8), referring expression comprehension (Figure 9), referring expression generation (Figure 9), semantic segmentation, (Figure 10) and referring expression segmentation (Figure 10).

# E   DISCUSSION WITH OOD QUESTIONS

We aim to investigate the generalization capability of our model, specifically its ability to handle out-of-distribution (OOD) questions that are not present in the training set. To this end, we design unconventional queries, as illustrated in Figure 11. For instance, our model correctly identifies the appendix as the smallest organ in a chest and abdomen CT scan—a concept not included in the training data. Similarly, when confronted with the grammatically unconventional query "smartest organ," the model appropriately responds with "Brain," despite this phrase not being part of the training data.

Our dataset includes questions describing anomalies, and we impose stricter constraints by limiting queries to one, three, and five words. Notably, our model successfully addresses these constrained queries, even though it was not explicitly trained for such scenarios. Moreover, when presented with queries related to surgical planning or seeking life advice, the model generates relevant responses, demonstrating its adaptability beyond the training data.

In summary, the M3D-LaMed model exhibits robust generalization capabilities for OOD problems. This proficiency is attributed to our approach of performing lightweight LoRA fine-tuning on the LLM rather than full-parameter fine-tuning, which preserves the LLM's original understanding and knowledge. By leveraging the inherent capabilities of the LLM and fine-tuning on new multi-modal datasets, our MLLM demonstrates enhanced professional and generalization capabilities. Consequently, developing a medical MLLM grounded in a robust LLM foundation proves to be indispensable.

# F   PROMPTS AND TEMPLATES

In our work, we frequently utilize various prompts and templates to guide LLM in different tasks, including data generation, model evaluation, and task instruction. Figures 12 and 13 depict prompts for data generation. Additionally, instruction templates for different tasks are provided in Figures 14, 15, 16, and 17. Figure 18 presents samples from our term dictionary.

| Image-text retrieval (IR) |
|---|

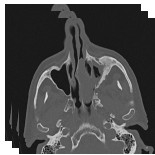

**Ground Truth & Rank 1**: **Frontal sinus completely opacified on left. Frontochoanal polyp. Small left maxillary sinus with mucosal thickening, bubbly secretion, and thickened outer walls.** Cavernous segment of ICA bilaterally exhibits extensively calcified atheromatous plaque.

**Rank 2**: **Marked mucosal thickening and total soft tissue opacification of maxillary, frontal and sphenoid sinuses as well as ethmoidal complexes merging with the nasal turbinates and causing partially obliteration of nasal cavities. Both osteomeatal complexes are obstructed.** Leftward nasal septal deviation is present.

**Rank 3**: **Markedly decreased size and near-complete opacification of both maxillary and sphenoid sinuses.** Symmetrical inward bowing of the anterior wall of the maxillary sinuses bilaterally. **Near-complete opacification of both frontal sinuses and ethmoid air cells with rarefaction of ethmoid bony septae. The ostio-meatal units, sphenoethmoidal recesses and frontal outflow tracts are compromised with thickened mucosa and retained secretions. Roomy nasal cavity with thickened mucosal and retained secretions.** The nasal septum is wavy with more bowing towards the right side.

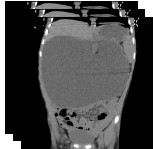

**Ground Truth & Rank-1**: **A 12x20x22cm cystic mass is seen within the abdominal cavity containing a focus of calcification and fat** associated with the pressure effect and displacement of adjacent abdominopelvic organs. The mass is not related to pelvic organs.

**Rank 2**: **Massive hepatomegaly which is largely replaced by cysts. No solid mass. Right kidney has multiple cysts without substantial enlargement or loss of cortex.** Left kidney is obstructed by proximal left ureteral stone with severe hydronephrosis, somewhat difficult to distinguish from the cortex which has been replaced by cysts.

**Rank 3**: **Very large heterogeneously enhancing retroperitoneal mass extending from the upper abdomen into the right pelvis displacing the bladder towards the left, filling much of the abdominal cavity.** Abdominal aorta and iliac vessels displaced and encased along the leftward aspect of the mass. **Obstruction of the right ureter by the mass with delayed right nephrogram.** Mild left hydronephrosis.

| Image-text retrieval (TR) |
|---|

**Gound Truth & Rank-1**: Diffuse thickening of the entire colon, mainly involving sigmoid and descending colon. Colonic dilatation with transverse colon diameter reaching 8.4 cm. No small bowel dilatation or pneumoperitoneum.

**GT & Rank 1** 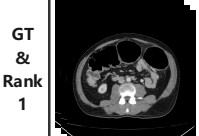

**Rank 2** 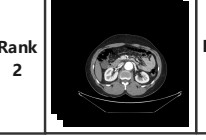 **Rank 3** 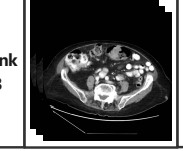 **Rank 3** 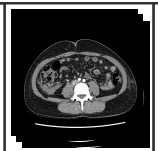 **Rank 4** 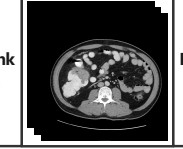 **Rank 5** 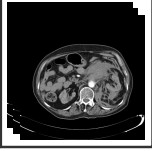

Figure 5: Qualitative comparisons on image-text retrieval. In each case, text with the same color represents identical content, while text with different colors signifies different content. The top-ranked samples exhibit similarities to ground truth (GT) in both image-to-text retrieval (IR) and text-to-image retrieval (TR) tasks.

| Report generation |
|---|
| 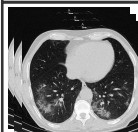 **\<Answer\>:** **Patchy ill-defined opacities with ground glass and crazy paving appearance are seen at both lungs particularly at subpleural regions.** |
| **Our:** **Patchy ill-defined ground glass opacities and consolidations are seen at both lungs predominantly at subpleural regions**

**RadFM:** CT chest with contrast showing hypoplastic right lung with venous drainage directly into IVC |
| 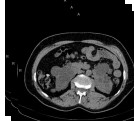 **\<Answer\>:** The lower poles of both kidneys are seen pointing medially and seen fused across the midline by an isthmus of renal tissue, with anteriorly mal-rotated hila denoting horseshoe kidney. **Multiple cysts are present at both kidneys.** The largest one on right and left sides measured 64mm and 61mm respectively. **A 5mm stone is observed at lower calyces of the right kidney.** Mild hydronephrosis is evident at right side. |
| **Our:** **A 20mm stone is present in the lower pole of the right kidney.** In addition, **a few non-enhanced simple cortical cysts are seen in both kidneys,** with maximum diameters of 20mm. Degenerative changes as osteophytosis are seen in the lumbar spine.Grade I spondylolisthesis of L5 on S1 is present with bilateral spondylolysis.

**RadFM:** CT,C+ portal venous phase |
| 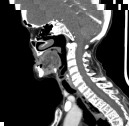 **\<Answer\>:** **A well-defined lesion seen at the inferior border of the left thyroid lobe, with reduced enhancement compared to the normal thyroid gland in the arterial phase but with greater washout than the thyroid gland in the delayed phase. The lesion measured about 16 x 12 mm with no calcification.** Diffuse decreased density of the imaged skeleton. |
| **Our:** **A hyperdense nodule measuring 9 x 8 mm is seen in the left thyroid lobe. The lesion is of low attenuation compared to the surrounding thyroid tissue and shows no enhancement on the post-contrast images.**

**RadFM:** Cystic lesion in the left sublingual space. |
| 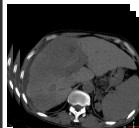 **\<Answer\>:** Triphasic CT of the liver demonstrates the following:Enlarged cirrhotic liver. **Multisegmental right hepatic lobe focals with faint homogenous wash-in of contrast in arterial phase and washout in venous and delayed phases.** Completely occluded of the portal vein lumen as well as its main branches with thrombosis. Mild abdominal ascites. Enlarged spleen. Small left kidney. |
| **Our:** **A large mass with enhancing central necrotic area is seen in the right hepatic lobe. The mass shows early enhancement and rapid washout in the portal venous phase.** The mass is abutting the right liver bed with no evidence of invasion.

**RadFM:** CT of the abdomen and pelvis demonstrates a large enhancing mass arising from the left kidney. The mass has central regions of hypodensity, presumably representing necrosis. The mass displaces the bowel, with no evidence of invasion. No nodal enlargement is seen. No calcification is evident. |

Figure 6: Qualitative comparisons with another method in report generation. Text with the same color indicates identical content, while different colors signify differing content. Our model exhibits superior performance to RadFM by generating more answer-identical content. We attempted to include GPT-4V in the comparison but encountered limitations, as it struggled to generate medical-related diagnostic recommendations.

Figure 7: Qualitative comparisons with other methods in closed-ended VQA. We compare methods across five types: plane, phase, organ, abnormality, and location, highlighting the superiority of our approach. In closed-ended VQA, GPT-4V sometimes relaxes the restrictions, because we provide choices that may turn medical advice into a multiple-choice question.

| Open-ended visual question answering |
|---|
|  **<Question>**: <Image> Which plane is displayed in the image?
**<Answer>:** Axial
**Our:** Axial
**RadFM:** The coronal plane.
**GPT-4V:** The images you provided are not of planes, but rather axial (transverse) slices from a CT (computed tomography) scan of a human body. |
|  **<Question>**: <Image> What is the CT phase shown in the image?
**<Answer>:** Portal venous phase
**Our:** Portal venous phase
**RadFM:** Portal venous phase.
**GPT-4V:** The CT phase shown in the images is the portal venous phase. |
|  **<Question>**: <Image> Which organ does the narrowing occur in?
**<Answer>:** Small bowel
**Our:** Small bowel
**RadFM:** In the splenic flexure.
**GPT-4V:** - |
|  **<Question>**: <Image> What is the abnormality seen in the splenic vein and artery?
**<Answer>:** Partial thrombosis
**Our:** Thrombosis
**RadFM:** There is an abnormal communication between the splenic vein and artery through a tortuous and dilated vessel.
**GPT-4V:** - |
|  **<Question>**: <Image>Which side of the heart is affected by the pericardial fluid?
**<Answer>:** Left side
**Our:** Left
**RadFM:** Right side.
**GPT-4V:** The pericardial fluid is surrounding all sides of the heart. It is not localized to one specific side but is present in the pericardial sac, which encases the entire heart. |

Figure 8: Qualitative comparisons with other methods in open-ended VQA. Similarly, our method demonstrates superior performance across five types. However, questions related to abnormality topics in open-ended VQA remain restricted by GPT-4V. In cases where no valid answer can be obtained, "-" is used to indicate this limitation.

Figure 9: Qualitative analysis on positioning tasks. We demonstrate two task forms: box output and box input, representing referring expression comprehension and referring expression generation, respectively. This demonstrates our model's effectiveness in completing the vision language positioning task. In the visualizations, the green box represents the ground truth, while the red box represents the prediction.

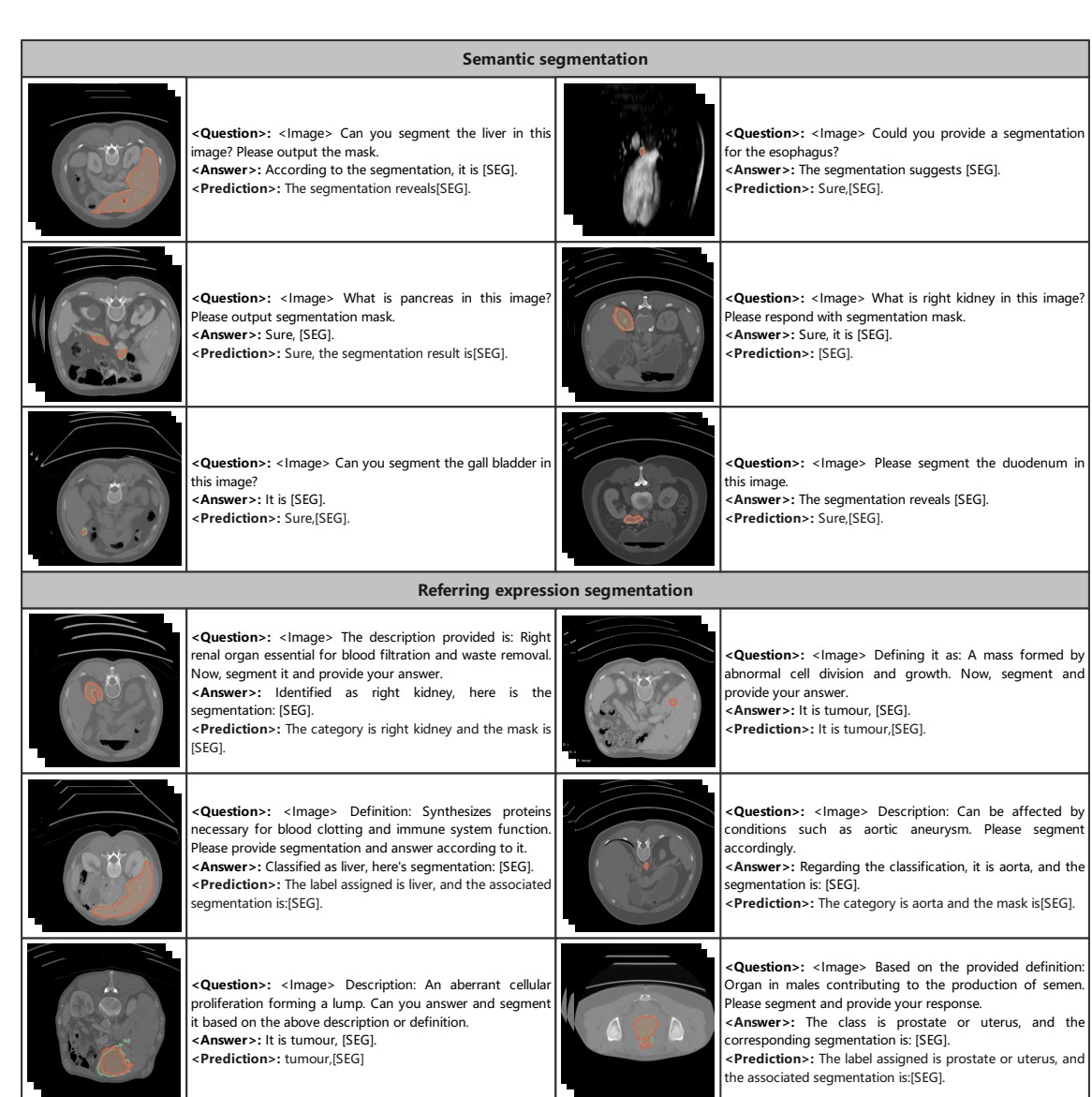

Figure 10: Qualitative analysis on segmentation tasks. We show two task forms: semantic segmentation and referring expression segmentation, highlighting our model's proficiency in segmentation tasks. In the visualizations, the green mask represents the ground truth, while the red mask represents the prediction.

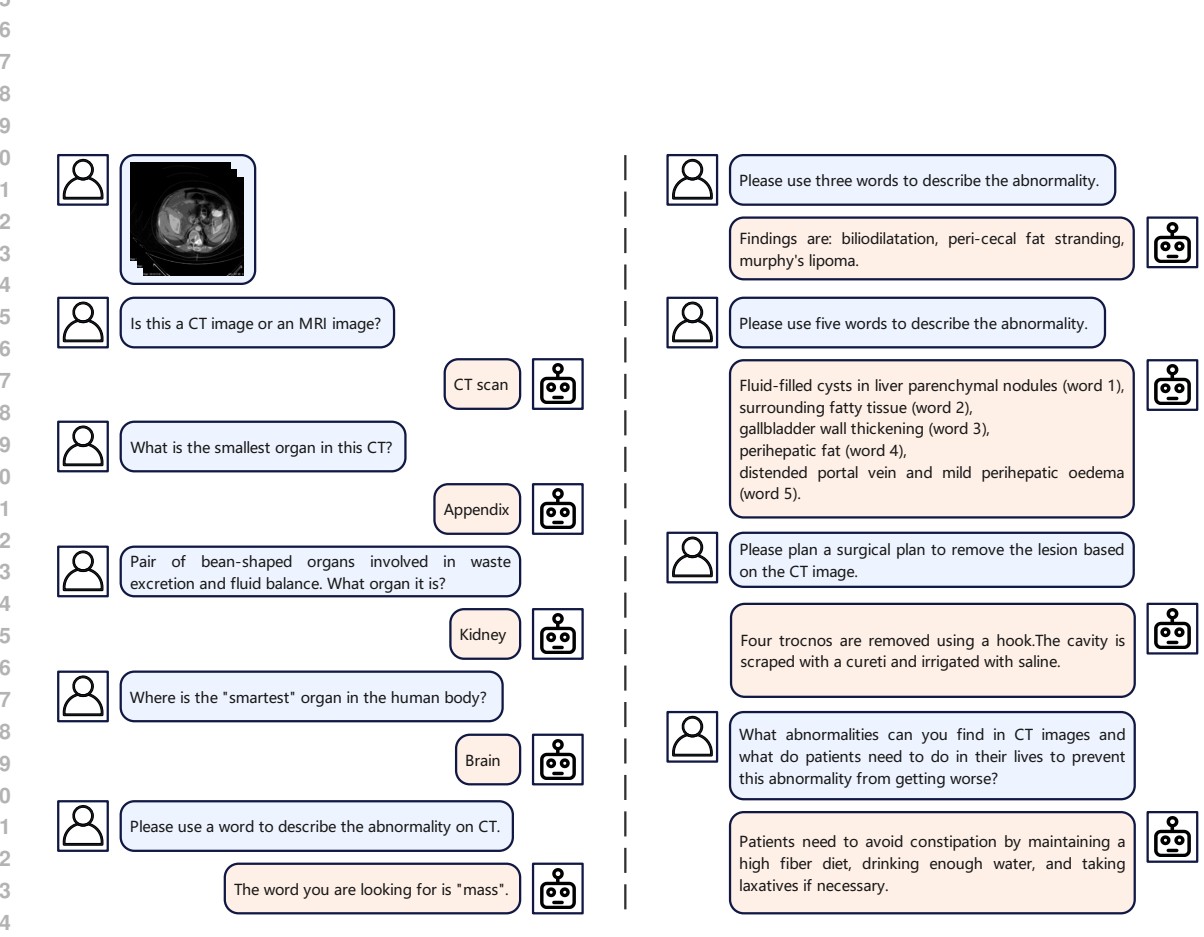

Figure 11: Case study on out-of-distribution (OOD) questions. We evaluate the M3D-LaMed model on OOD questions, where all queries are **NOT** related to the training data. Our findings indicate that M3D-LaMed demonstrates strong generalization capabilities, providing reasonable answers to OOD questions rather than producing nonsensical responses. In each conversation set, the avatar and questions on the left are provided by the user, while the avatar and answers on the right are generated by the M3D-LaMed model.

You are a medical AI visual assistant that can analyze a single CT image. You receive the file name of the CT image and the medical diagnosis report. The report describes multiple abnormal lesions in the image.

The task is to use the provided CT image and report information to create plausible 9 questions about the image. Each question corresponds to four options, and these questions come from the following 5 aspects:
1). Planes (axial, sagittal, coronal);
2). CT phase (non-contrast, contrast, arterial phase, portal venous phase, venous phase, delayed phase, parenchymal phase, renal cortical phase, dual phase, renal excretory phase, mixed arteriovenous, myelography, etc.) or window ( bone, lung, window, etc.);
3). Organ;
4). Abnormality type or description;
5). Abnormality position;

**Image:** $\{image\_file\_name\}$ # It provides basic information about planes and phase.
**Report:** $\{text\}$ # It provides detailed image findings and impressions.

**Desired format:**
1). Planes
Question-1: ...? Choice: A. ... B. ... C. ... D. ... Answer: A. ...
2). CT phase
Question-2: ...? Choice: A. ... B. ... C. ... D. ... Answer: A. ...
3). Organ
Question-3: ...? Choice: A. ... B. ... C. ... D. ... Answer: A. ...
4). Abnormality type or description
Question-4: ...? Choice: A. ... B. ... C. ... D. ... Answer: A. ...
Question-5: ...? Choice: A. ... B. ... C. ... D. ... Answer: A. ...
Question-6: ...? Choice: A. ... B. ... C. ... D. ... Answer: A. ...
5). Abnormality position
Question-7: ...? Choice: A. ... B. ... C. ... D. ... Answer: A. ...
Question-8: ...? Choice: A. ... B. ... C. ... D. ... Answer: A. ...
Question-9: ...? Choice: A. ... B. ... C. ... D. ... Answer: A. ...

Make the correct answers randomly distributed among the four choices.
If there is a true or false question, please ensure that the proportion of yes and no is equivalent. For example, Is ... ? Are ... ?, Do ... ?, Does ... ?, Did ... ?, Can ... ?.
Please do NOT ask directly what organs or abnormalities are visible in the image, as the answers are not unique. It would be best to use specific descriptions in your questions to ensure that other people can get an accurate answer even without providing choices.
Please be careful not to mention the file name and report. Always ask questions and answer as if directly looking at the image.

Figure 12: The prompt of VQA data generation. Specifically, we insert the image file name and report text into the placeholders ({}) within the prompt and feed it to LLM. Subsequently, we post-process the output of LLM to extract VQA data. Additionally, we observed that Qwen-72B (Bai et al., 2023) and ChatGPT (OpenAI, 2019) perform similarly in our data generation experiments, leading us to adopt the more cost-effective Qwen-72B model.

You are a medical AI visual assistant that can analyze a single CT image. Unfortunately you can't see the image but you can receive a diagnostic report of a local area in the CT image. The report describes the abnormal lesion in the image.

The task is to use the provided report information to create plausible 6 questions and answers about the image for reasoning segmentation tasks

**Report:** $\{text\}$ # It provides detailed image findings and impressions.

Questions and answers need to be structured from the report. But don't mention the report in Q&A. The question needs to be about a specific lesion area and requires segmentation of this area. The answer needs to use only one [SEG] symbol to refer to the segmentation area and provide a text explanation.

There are two types of questions: one type of question is answered and segmented based on description information, and the other type of question requires reasoning based on general and medical knowledge to obtain answers and segmentation.

**Example:**
1). Description-based
Question-1: Please segment where the liver cyst appears in the image. Answer: Sure, it is [SEG] on the upper right side of the liver.
2). Reasoning-based
Question-1: Can you segment the unusual part in this image and explain why? Answer: Sure, it is [SEG]. In the image, the unusual part is ...
Question-2: What can make the woman stand higher? Please output segmentation mask and explain why. Answer: Sure, [SEG]. The woman is standing higher by using ...
Question-3: If there are any lesions in the largest human body organ in the image, please segment them. Answer: The largest organ is the liver, where liver tumors are present, and the region is the [SEG].

**Desired output format:**
1). Description-based
Question-1: ...? Answer: ...
Question-2: ...? Answer: ...
Question-3: ...? Answer: ...
2). Reasoning-based
Question-4: ...? Answer: ...
Question-5: ...? Answer: ...
Question-6: ...? Answer: ...

Please construct a total of 6 sets of question and answer pairs according to the desired format, 3 sets of each type.
Using specific descriptions in your questions would ensure others can get an accurate answer.
Always ask questions and answer as if directly looking at the image.

Figure 13: The prompt of data generation for referring expression segmentation. Specifically, we insert the report description for a mask into the placeholders ({}) within the prompt and feed it to LLM. Subsequently, we post-process the output of LLM to extract instructions. This involves generating diverse descriptions and inferential questions from simple diagnostic reports. Qwen-72B is selected for this task due to its efficiency and performance.

**Report Generation:**

- Can you provide a caption consists of findings for this medical image?
- Describe the findings of the medical image you see.
- Please caption this medical scan with findings.
- What is the findings of this image?
- Describe this medical scan with findings.
- Please write a caption consists of findings for this image.
- Can you summarize with findings the images presented?
- Please caption this scan with findings.
- Please provide a caption consists of findings for this medical image.
- Can you provide a summary consists of findings of this radiograph?
- What are the findings presented in this medical scan?
- Please write a caption consists of findings for this scan.
- Can you provide a description consists of findings of this medical scan?
- Please caption this medical scan with findings.
- Can you provide a caption consists of findings for this medical scan?

Figure 14: Examples of instructions for report generation. These instructions typically include prompts or guidelines for generating specific sections or content within the medical reports. These instructions, along with corresponding images, are input into the MLLM together to facilitate the report generation process.

**Referring Expression Comprehension:**

Category Questions:

- Can you find the {} in this image? Give coordinates.
- Can you find {} in this image? Please output the coordinates.
- Please bounding the {} by box in this image.
- Where is {} in this image? Please respond with a bounding box.
- Where is {} in this image? Please output the box.
- Can you locate the {} in this image? Please output its coordinates.
- Could you mark the {} by bounding box in this image?
- Where can I find the {} in this image? Please provide its bounding box.
- Identify the indicated {} in this image. Please provide the coordinates of its bounding box.

Answers:

- Coordinates are {}.
- Sure, {}.
- Sure, it is {}.
- Sure, the bounding box is {}.
- {}.
- Here are the coordinates: {}.
- Of course, it's located at {}.
- The bounding box is given by {}.
- The box is {}.

Description Questions:

- Description: {} Please answer and find it by box based on the above description.
- Definition: {} Please answer and show the bounding box based on the above definition.
- Description: {} Can you answer and find it by coordinates based on the description?
- Definition: {} Please output the bounding box and answer based on the definition.
- Description: {} Respond and locate it using a bounding box according to the description.
- Definition: {} Please provide an answer and display the bounding box according to the given definition.
- Description: {} Can you identify and locate it by coordinates, following the provided description or definition?
- Definition: {} Please output the bounding box and provide an answer based on the provided definition.
- Based on the description or definition, please respond to {} and indicate its location with a bounding box.

Answers:

- The target is {} and the coordinates is {}.
- The category is {} and the bounding box is {}.
- It is {}, {}.
- {}, {}
- The target is identified as {} and its coordinates are {}.
- The category is {}, the bounding box is provided as {}.
- It is characterized by {}, with coordinates {}.
- The identified attributes are {}, {}.
- Describing it as {}, the corresponding box is {}.

Figure 15: Instruction templates for referring expression comprehension. These templates guide the construction of instruction data for the referring expression comprehension task. The data for this task is sourced from M3D-Seg, a segmentation dataset providing categories and bounding boxes. In category questions, categories are inserted into question templates' placeholders ({}) as input, while bounding boxes are inserted into answer templates' placeholders ({}) as output. In description questions, categories are converted into descriptions using the term dictionary. These instruction templates facilitate the generation of instruction data for referring expression comprehension.

**Referring Expression Generation:**

Category Questions:

- What target is present within the coordinates {} ?
- Does the bounding box {} contain any target?
- Within the specified region {}, what target is present?
- Do you know what it is in the bounding box {}?
- What is it in this region {}?
- What object is located within the coordinates {}?
- Within the specified area {}, what object can be found?
- Can you identify the object within the bounding box {}?
- What object is present in this region {}?

Answer:

- The target is {}.
- Sure, the bounding box contains {}.
- Sure, it is {}.
- Sure, {} is in the bounding box.
- {}.
- The object is {}.
- Of course, it's {}.
- Certainly, {} can be found in the bounding box.
- Yes, the bounding box includes {}.

Description Questions:

- Please describe the target and its function based on the box {} in the image.
- Do you know what is it in this bounding box {}? Answer and explain it.
- What's the target in the bounding box {}? What function does it have?
- What is the area marked with a box {} in the image? Can you explain it?
- Could you describe the object and its purpose within the bounding box {} in the image?
- Can you identify and describe the object within this bounding box {}? Please explain.
- What is the object located in the bounding box {}? Could you explain its function?
- Could you describe the area outlined by the box {} in the image? Please explain its significance.

Answer:

- Sure, it is {}. {}.
- The category is {}. {}.
- It is {}, {}.
- {}, {}
- The target is identified as {} and its description is {}.
- The category is {}. Description: {}.
- It is characterized by {}, {}.
- The identified attributes are {}, {}.
- Sure, it is {}. Describing it as {}.

Figure 16: Instruction templates for referring expression generation. These templates facilitate the construction of instruction data for the referring expression generation task. In category questions, bounding boxes are inserted into question templates' placeholders ({}) as input, while categories are inserted into answer templates' placeholders ({}) as output. Similarly, in description questions, categories are converted into descriptions using the term dictionary. The model is expected to output both the target and its description as answers.

**Semantic Segmentation:**

Question:

- Can you segment the {} in this image?
- Can you segment {} in this image? Please output the mask.
- Please segment the {} in this image.
- What is {} in this image? Please respond with segmentation mask.
- What is {} in this image? Please output segmentation mask.
- Could you provide a segmentation for the {}?
- Segment {} from this image and provide the mask, please.
- Please provide a segmentation mask for the {} in this image.
- Can you identify and segment the {} in this image?

Answer:

- It is [SEG].
- Sure, [SEG].
- Sure, it is [SEG].
- Sure, the segmentation result is [SEG].
- The segmentation indicates [SEG].
- According to the segmentation, it is [SEG].
- The segmentation reveals [SEG].
- The segmentation suggests [SEG].
- From the segmentation, it appears to be [SEG].

**Referring Expression Segmentation:**

Question:

- Description: {} Please answer and segment based on the above description.
- Definition: {} Please answer and segment based on the above definition.
- Description: {} Can you answer and segment it based on the above description or definition.
- Definition: {} Please output segmentation mask and answer based on the above description or definition.
- Provided description: {} Please segment accordingly.
- Given definition: {} Please provide segmentation and answer according to it.
- The description provided is: {} Now, segment it and provide your answer.
- Based on the provided definition: {} Please segment and provide your response.
- Describing the object as: {} Can you segment it accordingly?

Answer:

- The target is {} and the segmentation mask is [SEG].
- The category is {} and the mask is [SEG].
- It is {}, [SEG].
- Identified as {}, here is the segmentation: [SEG].
- Categorized as {}, the segmentation is: [SEG].
- The class is {}, and the corresponding segmentation is: [SEG].
- Regarding the classification, it is {}, and the segmentation is: [SEG].
- Classified as {}, here's the segmentation: [SEG].

Figure 17: Instruction templates for segmentation tasks. In semantic segmentation, categories are inserted into question templates' placeholders ({}) as input. For referring expression segmentation, descriptions are inserted into question templates' placeholders ({}) as input. In both cases, all answers include a special token [SEG], which instructs the segmentation module. This token is crucial for guiding the segmentation process based on the provided input.

```
{
  "liver": [
      "Primary organ responsible for detoxifying the blood by removing harmful substances.",
      "Produces bile, a fluid that aids in the digestion and absorption of fats.",
      "Stores and regulates glycogen, a crucial energy reserve for the body.",
      "Synthesizes proteins necessary for blood clotting and immune system function.",
      "Plays a central role in metabolism, including the breakdown of carbohydrates and fats.",
      "Large organ in the upper right abdomen with various metabolic functions.",
      ......],
  "left lung": [
      "Organ located on the left side of the chest involved in respiration.",
      "Respiratory organ situated in the left thoracic cavity.",
      "Lung found on the left side of the body responsible for breathing.",
      "Pulmonary structure on the left side of the chest responsible for gas exchange.",
      "Left-sided respiratory organ essential for oxygen exchange.",
      "Organ situated in the left thorax responsible for oxygenating blood.",
      "Lung located in the left hemithorax involved in ventilation.",
      ......],
  "kidney": [
      "Pair of organs responsible for filtering waste from the blood.",
      "Organ duo involved in removing waste and excess fluids from the body.",
      "Pair of bean-shaped organs essential for regulating bodily fluids.",
      "Organs crucial for filtering blood and producing urine.",
      "Pair of vital organs filtering blood and maintaining fluid balance.",
      "Bean-shaped organs integral to waste removal and urine production.",
      "Organs vital for removing toxins and excess fluids from the body.",
      ......],
  "heart": [
      "Organ responsible for pumping blood throughout the body.",
      "Muscular organ that circulates blood throughout the circulatory system.",
      "Vital organ that pumps oxygenated blood to tissues and organs.",
      "Primary pump of the circulatory system, supplying oxygen to tissues.",
      "Central organ of the cardiovascular system, propelling blood throughout the body.",
      "Main organ of the circulatory system, distributing nutrients and oxygen.",
      ......],
  "liver tumor": [
      "Abnormal growth in liver tissue.",
      "Mass of cells forming in the liver.",
      "Neoplastic lesion found in the liver.",
      "Pathological growth occurring in liver tissue.",
      "Uncontrolled cell proliferation in the liver.",
      "Anomaly of tissue growth within the liver.",
      ......],
  ......
}
```

Figure 18: Examples from the term dictionary. The term dictionary contains multiple descriptions for each medical term. These descriptions are generated through ChatGPT. With numerous medical terms included, this dictionary is crucial in transforming semantic categories into detailed descriptions. These descriptions are essential for facilitating positioning and segmentation tasks.

