# OpenReview forum: "M3D: Advancing 3D Medical Image Analysis with Multi-Modal Large Language Models"
_ICLR.cc/2025/Conference — ICLR 2025 Conference Withdrawn Submission_

### Official Review · Reviewer_fCBf · 2024-10-25

**Soundness:** 2
**Presentation:** 3
**Contribution:** 2
**Rating:** 3
**Confidence:** 5

**Summary:**

This paper presents the development of a 3D vision-language model focusing on the interpretation of 3D computed tomography imaging. A CLIP-guided multi-modal pretraining approach is proposed and evaluations are performed across a variety of clinically relevant and adjacent tasks. Some relevant baselines are included. No trained models or source code is provided.

**Strengths:**

*	Vision-language pretraining for 3D CT scans can be computationally prohibitive. This paper tackles this approach using a novel 3D ViT and perceiver-based approach. This allows for good image-text alignment.
*	The proposed model has a nice unified training paradigm and interface across tasks
*	A variety of tasks are evaluated across task types, including VQA, segmentation, report generation, etc. Tasks like report generation and report generation currently have mid-to-high level clinical importance.
*	Multiple examples of task performance is provided in appendix

**Weaknesses:**

*	For the training of their proposed model, the authors used the Radiopedia dataset. However, radiopedia explicitly prohibits scraping the data from their website for such uses. https://radiopaedia.org/terms . It is mentioned that "Radiopaedia is not designed or intended to be used as an imaging data-set for Machine Learning/Artificial Intelligence research, nor is it available for bulk download using an automated script or robot or any other process whereby many cases are downloaded." Copyrights to data is clearly a very active topic currently, and it is problematic that the model developed here does not abide by the original copyrights of a crowd-compiled dataset. Overall, it does not seem that the authors have the ability to download and freely distribute the dataset as claimed.
*	No comparisons are performed with more relevant task specific models or more stat of the art 3D vision language models. From a baselines perspective, it would be beneficial to compare to the BiomedCLIP model for retrieval as well as training. For segmentation, using more common models like nnunet or swinunetr would be more fair. For general 3D VL training, there is no comparison to a couple of recent papers that are very related: 1. Hamamci et al. Developing Generalist Foundation Models from a Multimodal Dataset for 3D Computed Tomography. And  2. Blankemeier et al. Merlin: A Vision Language Foundation Model for 3D Computed Tomography
*	The M3D-RefSeg is the TotalSegmentator dataset. All that was done as a mapping between class number and the text description. Rebranding it as a unique contribution is not appropriate.
*	Table 1 describes a Texts column - it is unclear what this is. Tokens? Also, referring to segmentation masks seems unlikely to be considered as text since its a specific class anyway at the end of the day.
*	CLIP struggles with long context and fine grained learning. It does not seem that there was any additional method used besides the traditional CLIP in this study, which may be problematic because of the high density of text information and low variability of the images.
*	It is unclear to me what the value of the benchmarking positioning task is, when one could simple perform the segmentation for the same tissues anyway? Since the bounding box is generated from the segmentation label, there is limited new value provided.

**Questions:**

*	How was noisy VQA data eliminated? What was the rubric and evaluation criteria? The data quality did not include any analysis of open ended questions, which is likely of most importance of all tasks. Further, who were the experts used? Board certified radiologists?
*	Did interpolating the images to a lower resolution hurt performance of the model? This is a relevant ablation study.
*	What metric is shown in Table 4?
*	It is not clear why natural language generation metrics should be used to assess open ended VQA when the correct answers are available in the closed form solution.

**Details Of Ethics Concerns:**

For the training of their proposed model, the authors used the Radiopedia dataset. However, radiopedia explicitly prohibits scraping the data from their website for such uses. https://radiopaedia.org/terms . It is mentioned that "Radiopaedia is not designed or intended to be used as an imaging data-set for Machine Learning/Artificial Intelligence research, nor is it available for bulk download using an automated script or robot or any other process whereby many cases are downloaded."

Copyrights to data is clearly a very active topic currently, and it is problematic that the model developed here does not abide by the original copyrights of a crowd-compiled dataset. Overall, it does not seem that the authors have the ability to download and freely distribute the dataset as claimed.

---

### Official Review · Reviewer_zxW4 · 2024-11-01

**Soundness:** 3
**Presentation:** 3
**Contribution:** 3
**Rating:** 6
**Confidence:** 4

**Summary:**

The paper introduces M3D-LaMed, a generalist large language model designed for comprehensive 3D medical image analysis. The study tackles challenges in 3D medical imaging, such as data scarcity and high computational costs, which previously limited exploration in this field. M3D-LaMed is specialized for eight critical tasks, including image-text retrieval, report generation, visual question answering (VQA), positioning, and segmentation.

Key contributions of the paper include:
M3D-LaMed Model: A multi-modal large language model (MLLM) specifically designed for various 3D medical imaging tasks.
M3D-Data: The largest public 3D multi-modal medical dataset to date, containing 120,000 image-text pairs and 662,000 instruction-response pairs, covering a range of diseases and tasks.
M3D-Bench: The first 3D multi-modal benchmark that evaluates performance across eight 3D medical tasks, with specific metrics for image-text retrieval, VQA, positioning, and segmentation.
The results demonstrate that M3D-LaMed surpasses existing models across these tasks, establishing a robust framework for 3D medical image analysis and promoting further research in medical imaging applications​

**Strengths:**

M3D-LaMed represents a highly original application of multi-modal large language models (MLLMs) to the under-explored domain of 3D medical imaging. By moving beyond traditional 2D image analysis, it highlights the added value of 3D data in medical diagnostics, addressing a significant limitation in prior work.

The introduction of a 3D spatial pooling perceiver to efficiently process 3D tokens while maintaining spatial information is a novel approach that creatively adapts existing methods to suit the unique demands of 3D medical data.

M3D-LaMed’s generalist approach and performance across multiple complex tasks make it a significant contribution to the field of medical imaging, as it offers a versatile tool adaptable to various clinical applications.

The publicly available M3D-Data and M3D-Bench will serve as valuable resources for the broader community, facilitating the development and evaluation of future models in 3D medical image analysis.

By addressing spatial and contextual complexity in 3D images, M3D-LaMed enhances diagnostic capabilities, which could translate into better clinical decision-making and improved patient outcomes.

**Weaknesses:**

Lack of Explanation for Resizing Method: The authors mention using Min-Max normalization and resizing 3D CT images to a standard dimension of 32×256×256, but the specific resizing technique used (e.g., interpolation method, aspect ratio considerations) is not clarified. Without this information, it is difficult to assess how the resizing might affect spatial integrity in 3D images, especially for subtle structures critical in medical analysis.
No Comparative Study on Resizing: It would be beneficial if the authors conducted a comparative study to evaluate the effects of different resizing methods on performance, as variations in preprocessing can impact model accuracy on medical data.

Lack of Comparative Analysis with MLP: The paper introduces a 3D Perceiver module to handle token reduction for efficient 3D processing, but it does not compare this approach directly against a simpler alternative, such as a Multi-Layer Perceptron (MLP). A controlled comparison would demonstrate the added value of the 3D Perceiver and validate its effectiveness in preserving spatial information while reducing computation.
No Ablation Study on the 3D Perceiver: Including an ablation study that isolates the impact of the 3D Perceiver module would strengthen the paper by empirically verifying its contribution to the model’s performance.

Limited Comparative Analysis with 2D Models: Although the paper focuses on 3D models, a more detailed comparison with state-of-the-art 2D models on similar tasks (e.g., segmentation or retrieval) would highlight the advantages and limitations of the 3D approach. By doing so, the authors could clarify whether the shift to 3D improves performance in specific areas, such as spatial recognition, and where it might face challenges, such as increased computational costs.

Lack of Discussion on Demographic and Disease Variety: While M3D-Data is substantial in size, the paper does not address whether it includes diverse demographic groups or a wide range of diseases, which are crucial for generalizable models in healthcare. An analysis of the dataset’s coverage across demographic and pathological categories would provide insight into its representativeness and the potential for biases.
Absence of Evaluation for Dataset Representativeness: Ensuring that M3D-Data includes representative samples across age, gender, and disease types would enhance the model’s applicability. Without this, there’s a risk that M3D-LaMed may perform less effectively on underrepresented groups, limiting its utility in real-world clinical applications.

**Questions:**

Could you specify the resizing method used to preprocess 3D CT images to the standard dimension of 32×256×256? For example, was a specific interpolation technique applied, or were aspect ratios adjusted?

Did you compare the 3D Perceiver module directly with a simpler alternative, like an MLP, to assess its contribution to token reduction and spatial information preservation?

Although the focus is on 3D image analysis, did you evaluate M3D-LaMed’s performance against state-of-the-art 2D models on similar tasks to establish a clearer benchmark for the benefits and limitations of 3D approaches?

Could you provide more detail on the demographic diversity (e.g., age, gender, ethnicity) and disease types included in M3D-Data? Was any effort made to ensure the dataset’s representativeness across these dimensions?

M3D-Data primarily consists of CT images. Did you consider evaluating M3D-LaMed’s performance across other 3D imaging modalities, such as MRI or PET, to assess the model’s robustness in handling different data types?

While Dice score is used to measure segmentation performance, did you evaluate M3D-LaMed’s segmentation accuracy with respect to clinically relevant boundaries or subtle anatomical variations?

Could you provide details on the computational requirements for deploying M3D-LaMed in clinical settings? How do its processing times compare to those of 2D models, and what are the implications for clinical feasibility?

Could you clarify the licensing and accessibility of M3D-Data and M3D-Bench? Will these resources be publicly available to the community, and if so, under what conditions?

Given that M3D-LaMed still faces challenges with high-resolution 3D data, do you foresee any future improvements to the model architecture or data processing to better handle long token sequences in 3D?

---

### Official Review · Reviewer_eHtN · 2024-11-02

**Soundness:** 3
**Presentation:** 3
**Contribution:** 3
**Rating:** 6
**Confidence:** 3

**Summary:**

The paper proposes M3D-LaMed, a general model tailored for 3D medical image analysis. It includes a specialized 3D vision encoder, a 3D spatial pooling perceiver, and a large language model (LLM). Moreover, the authors propose M3D-Data, a  3D medical image-text pairs that can be used for several tasks i.e segmentation,  report generation, VQA, positioning, and image-text retrieval.  Comparisons with baselines models (PMC-CLIP and RadFM) reveal the superiority of M3D-LaMed.

**Strengths:**

•	M3D-Data is an interesting contribution to the community. This large-scale dataset enables further research into 3D vision and language for medical applications.
•	The addition of the spatial pooling perceiver in the 3D image encoder proves performant.
•	Evaluation: M3D-LaMed is thoroughly evaluated across several tasks strengthening the validity and usefulness of the current architecture. Several LLM evaluation metrics are used to quantitatively assess the results

**Weaknesses:**

•	Computational Efficiency: handling 3D images and text is known to be computationally demanding. Authors could have developed more on the problems that could arise if using longer text sequences or higher resolution images.
•	How well M3D-LaMed works on other medical imaging modalities?
•	The architecture of M3D-LaMed includes a 3D image encoder, a perceiver, and LLM but the interaction and importance of those three components is not fully explained.

**Questions:**

•	How each of the three components in M3D-LaMed contribute to performance?
•	Could your model include real-word metrics and how could we assess its impact in clinical settings?
•	How generalisable is the current model to out-of-domain data, and other modalities such as MRI and ultrasound?

---

### Official Review · Reviewer_9svC · 2024-11-03

**Soundness:** 3
**Presentation:** 3
**Contribution:** 3
**Rating:** 6
**Confidence:** 4

**Summary:**

Due to the expensive training cost and the limited of text-labeled data, limited research have been proposed to train a 3D medical image model direct for analysis, and more focused on 2D image slices. Therefore, this paper proposed an interesting idea to tackle 3D medical image and has three main contributions:
1) Proposed a general medical foundation model M3D-LaMed that can handle 8 3D medical task, including image-text retrieval, report generation, visual question answering, positing and segmentation.
2) Proposed a new dataset for public usage with 120K image-text and 662K instruction-response task
3) Performed extensive benchmarking with the current foundation models and the proposed generalist model

**Strengths:**

The strength of this paper can be concluded as follows:
1) Significant contribution on creating a public dataset for different tasks, as there is limited datasets that can be used in public, especially referring expression segmentation and image-text retrieval
2) This paper proposes a new model completely focused on 3D and demonstrate the success of using simple way to adapt 3D features into large language models (LLMs) for finetuning with LORA, which shows an interesting contribution on how to link 3D features with LLMs
3) The generalist model demonstrates a great variety on performing different tasks, showing a powerful generalizability of the learned semantics inside the foundation model

**Weaknesses:**

The weakness of this paper can be concluded as follows:
1) As the model is focused on using 3D image analysis, there is a lot of 3D foundation backbone proposed such as 3D swin transformer, 3D large kernel convolution, there is limited discussion in this paper on why it is important to use ViT here.
2) As the paper benchmarks the performance across different models, only a few baselines comparisons is performed with the adjustment on LLMs and one model with respect to the corresponding task.

**Questions:**

This is a really interesting paper and I think it is a great contribution on how to adapt 3D medical image features into LLMs, but I have a few concerns on the benchmarking part:
1) For image-text retrieval task, you only perform PMC-CLIP as one of your baselines. However, it should demonstrate your 3D model's performance is outperformed 2D aggregation baselines then we should move on to 3D. There is still a lot of embedding generating approach and use embeddings to perform image-to-text / text-to-image retrieval, such as:
- Zhang, Sheng, et al. "BiomedCLIP: a multimodal biomedical foundation model pretrained from fifteen million scientific image-text pairs." arXiv preprint arXiv:2303.00915 (2023).
- Zhang, Kai, et al. "Biomedgpt: A unified and generalist biomedical generative pre-trained transformer for vision, language, and multimodal tasks." arXiv e-prints (2023): arXiv-2305.

2) For 3D segmentation task, you also only perform SegVol as one your segmentation baseline. Still, there is a lot of 3D baselines out there e.g. SwinUNETR, 3D UX-Net, nnUNet. Also, as you have done pretraining with CLIP, you should also demonstrate that your finetuned model with LLM is better than purely supervised models. Right now, your experiments for Table 6 is like ablation studies, changing different LLMs to see the difference in segmentation performance and search for the best LLMs. It will be great if you can add some of your experiments into this paper and their code is already public for usage:
- Hatamizadeh, Ali, et al. "Swin unetr: Swin transformers for semantic segmentation of brain tumors in mri images." International MICCAI brainlesion workshop. Cham: Springer International Publishing, 2021.
- Lee, Ho Hin, et al. "3d ux-net: A large kernel volumetric convnet modernizing hierarchical transformer for medical image segmentation." arXiv preprint arXiv:2209.15076 (2022).
- Isensee, Fabian, et al. "nnU-Net: a self-configuring method for deep learning-based biomedical image segmentation." Nature methods 18.2 (2021): 203-211.

---

### Note · Authors · 2024-11-15

**Comment:**

Dear Reviewers and AC,

Thank you very much for your constructive comments! We are glad to see that three reviewers consistently rate our paper "good" in all metrics of Soundness, Presentation and Contribution. However,  Reviewer fCBf  is absolutely certain of "reject, not good enough". Hence, we decided to withdraw this submission.

Thank you again for your valuable time!

Authors

**Withdrawal Confirmation:**

I have read and agree with the venue's withdrawal policy on behalf of myself and my co-authors.